Effects of semaglutide on gut microbiota, cognitive function and inflammation in obese mice

Feng Jing 1 2
Teng Zhenjie 3 4
Yang Yu 1
Liu Jingzhen 2
Chen Shuchun chenshuc2014@163.com 1 2
1 Department of Endocrinology, Hebei Medical University , Shijiazhuang , China
2 Department of Endocrinology, Hebei General Hospital , Shijiazhuang , China
3 Department of Neurology, Hebei Medical University , Shijiazhuang , China
4 Department of Neurology, Hebei General Hospital , Shijiazhuang , China
Sotelo-Mundo Rogerio
Electronic publication date: 2024 Aug 12
Publication date: 2024
Volume: 12
Electronic Location ID: e17891
Received 2024 Mar 3; Accepted 2024 Jul 18
Copyright: ©2024 Feng et al.
Copyright year: 2024
Copyright holder: Feng et al.
License: This is an open access article distributed under the terms of the Creative Commons Attribution License, which permits unrestricted use, distribution, reproduction and adaptation in any medium and for any purpose provided that it is properly attributed. For attribution, the original author(s), title, publication source (PeerJ) and either DOI or URL of the article must be cited.
License URL: https://creativecommons.org/licenses/by/4.0/

Keywords: Glucagon-like peptide-1 receptor agonist, Semaglutide, Gut microbiota, Obesity, Cognitive function

Funding: The Hebei Provincial Central Leading Local Science and Technology Development Funds Project 206Z7702G This study was supported by the Hebei Provincial Central Leading Local Science and Technology Development Funds Project (206Z7702G). The funders had no role in study design, data collection and analysis, decision to publish, or preparation of the manuscript.

==============================
Objective

This study aims to investigate the effects of semaglutide on gut microbiota, cognitive function, and inflammation in obese mice.

Method

Twenty-four C57BL/6J male mice were randomly assigned to three groups: a normal-chow diet group (NCD, n = 8), high-fat diet group (HFD, n = 8), and HFD+semaglutide group (Sema, n = 8). The mice were fed a HFD to establish an animal model of obesity and then administered with semaglutide or saline for 12 weeks. Cognitive function was assessed using the Morris water maze test. Serum pro-inflammatory cytokines were measured. 16S rRNA gene sequencing technology was used to explore gut microbiota characteristics in obese mice.

Result

Obese mice showed significant cognitive impairment and inflammation. Semaglutide improved cognitive function and attenuated inflammation induced by a HFD diet. The abundance of gut microbiota was significantly changed in the HFD group, including decreased Akkermansia, Muribaculaceae, Coriobacteriaceae_UCG_002, Clostridia_UCG_014 and increased Romboutsia, Dubosiella, Enterorhabdus. Whereas semaglutide could dramatically reverse the relative abundance of these gut microbiota. Correlation analysis suggested that cognitive function was positively correlated with Muribaculaceae and Clostridia_UCG_014, and negatively associated with Romboutsia and Dubosiella. Romboutsia was positively correlated with TNFα, IL-6 and IL-1β. While Clostridia_UCG_014 was negatively related to TNFα, IL-6 and IL-1β.

Conclusions

For the first time semaglutide displayed different regulatory effects on HFD-induced gut microbiota dysbiosis. Semaglutide could regulate the structure and composition of gut microbiota associated with cognitive function and inflammation. Thus, affecting gut microbiota might be a potential mechanism of semaglutide in attenuating cognitive function and inflammation.

Introduction

In China, the obesity prevalence has grown dramatically since the 1980s (Mu et al., 2021), and has reached alarming levels. It is predicted that one billion people globally, including one in five women and one in seven men, will be living with obesity by 2030 (World Obesity Federation, 2022). Obesity has become a global pandemic and is a heavy medical, economic, and social challenge for patients and their families. Obesity is associated with hypertension, dyslipidemia, and insulin resistance, acting as a main contributor to cardiovascular morbidity and mortality. More recently, obesity has also been identified as a risk factor for the development of a wide range of neurological disorders (O’brien et al., 2017). A recent study has shown that patients with severe obesity are more likely to have severe cognitive impairment (Zhang et al., 2022a; Zhang et al., 2022b). Obese children and adolescents, as well as those with metabolic syndrome, display lower cognitive function, further distinguishing obesity-associated mild cognitive impairment from age-related dementia (Yau et al., 2012; Liang et al., 2014). Furthermore, strategies to reduce obesity in childhood could contribute to improvements in cognitive performance in midlife (Tait et al., 2022). Consistent with clinical data, high-fat diet feeding alters hippocampal structure and function in animal models (Jurdak, Lichtenstein & Kanarek, 2008; Kosari et al., 2012). For example, plasma membrane association of the insulin-sensitive glucose transporter, GLUT4, was reduced in the hippocampus of obese rats (Winocur et al., 2005). Besides, inflammation and insulin resistance provide insight into the mechanisms underlying obesity-induced cerebral changes (Maric, Woodside & Luheshi, 2014; Niepoetter et al., 2021). There is great significance in exploring the risk factors and preventative measures of obesity-related cognitive impairments.

Over the past decade, many studies have shown that gut microbiota is critical for maintaining human physiology, metabolism, and immunity (Zhang et al., 2022a; Zhang et al., 2022b). Microbial imbalances have been linked to various associated diseases, including obesity (Tait et al., 2022), diabetes (Zhou et al., 2022), NAFLD (Moreira et al., 2018), coronary heart disease (Liu et al., 2020) and psychiatric disorders (Nguyen et al., 2021). An increasing body of evidence has emphasized that obesity is associated with an altered composition of gut microbiota (Thingholm et al., 2019) and the gut microbial markers of obesity may be useful in improving psychological and metabolic health. A study has revealed that Akkermansia muciniphila improves cognitive function in aged mice by modulating inflammation-related pathways and reducing levels of the pro-inflammatory cytokine IL-6 (Zhu et al., 2023). In a cross-sectional study, microbial community composition is associated with domain-specific and global measures of cognition (Meyer et al., 2022). Further, GLP-1 receptor agonists are known to affect the intestinal immune system and change the gut microbiota (Zhao et al., 2018; Charpentier et al., 2021). There is evidence that liraglutide could regulate the composition of the gut microbiota in HFD-fed mice, specifically increasing the abundance of Akkermansia (Zhao et al., 2022). Although semaglutide has been reported to exert an anti-obesity effect in various studies, the effects on the gut microbiota in obese mice have not been fully elucidated. Whether semaglutide could help increase the beneficial gut microbiota of HFD-fed mice, which might have impacts on improving cognitive function and inflammation, is poorly understood.

To investigate the underlying mechanisms of semaglutide, we used 16S rDNA sequencing to analyze gut microbiota composition. Spearman’s correlation analysis was performed to explore the relationship between inflammatory markers or cognitive function and gut microbiota composition.

Materials & Methods

Animals and experimental design

A total of 24 male specific-pathogen-free grade C57BL/6J mice (6-week-old) were obtained from Hebei In vivo Biotechnology Co., Ltd (Hebei, China) and acclimated for 1 week. Animal care was carried out according to established guidelines. The mice were housed in ventilated cages (485 ×200 ×200 mm; 4 mice per cage; temperature: 20−24 °C; humidity: 55% ±10%). Thereafter, they were randomly divided into a normal-chow diet group (NCD, n = 8) or high-fat diet group (HFD, n = 16). The HFD group was administered a HFD (20% carbohydrate, 20% protein, and 60% fat) for 12 weeks to induce obesity in mice. The HFD group was further assigned into two groups according to a randomized block design: the HFD group (n = 8) and the Sema group (HFD + semaglutide, n = 8). The Sema group received a daily subcutaneous injection of semaglutide (Bagsværd, Novo Nordisk, Denmark, 30 nmol/kg/d), while the NCD and HFD groups were injected with equal volumes of saline for a further 12 weeks. The weight of all mice was measured after the intervention. The Morris water maze test assessed cognitive performance in each group (n = 8). Fecal and serum samples were collected for 16S rRNA sequencing (n = 6) and cytokines measurement (n = 6). Feces were collected in sterile cryopreservation tubes and then labeled and stored in a −80 °C refrigerator. At the end of the study, all mice were anesthetized with 1% sodium pentobarbital solution (50 mg/kg, intraperitoneal injection). This study was approved by the Animal Ethics Committee of Hebei General Hospital (202332), and every effort was made to minimize animal suffering.

Measurement of plasma cytokines

Serum samples were collected from the retro-orbital sinus after the mice received an intraperitoneal injection of pentobarbital sodium. Blood was centrifuged at 3000 × g for 10 min at 4 °C to obtain plasma for measurement of cytokines such as tumor necrosis factor (TNF) α, IL-6, and IL-1β. Cytokine concentrations are expressed in pg/mL.

Morris water maze test

The Morris water maze (MWM) test assessed spatial memory in all mice. The MWM test was conducted in a circular pool (diameter 120 cm, height 45 cm) filled with water at 24 to 26 °C. A visible escape platform (diameter 11 cm) was submerged approximately 1.5 cm under the water surface. In each acquisition trial, the mice started from one of the four quadrants by facing the wall of the tank. Each mouse was trained to reach the platform within 60 s for five consecutive days. If a mouse did not find the platform, it was gently guided to the platform and allowed to stay for 10 s. On the sixth day, the platform was removed and all mice were subjected to probe trials. Mice were allowed to swim in the pool for 60 s. Data were collected using a computerized animal tracking system (Shanghai Jiliang Software Science & Technology Co., Ltd, Shanghai, China), which recorded the path length, swimming speed, the time and latency to reach the platform and the number of platform crossings.

Gut microbiota analysis

Fecal DNA was extracted using the OMEGA Soil DNA Kit (D5625-01) (Omega Bio-Tek, Norcross, GA, USA) and subjected to PCR amplification using specific primers targeting the V3 and V4 hypervariable regions of the 16S rRNA gene. The specific forward primer was 341F 5′-CCTAYGGGRBGCASCAG-3′, and the reverse primer was 806R 5′-GGACTACNNGGGTATCTAAT-3′. Thermal cycling consisted of initial denaturation at 98 °C for 1 min, 30 cycles of denaturation at 98 °C for 10 s, annealing at 50 °C for 30 s, and elongation at 72 °C for 30 s. Finally, 72 °C for 5 min.

The amplified PCR products were extracted by 2% agarose gel electrophoresis and purified by Quant-iT PicoGreen dsDNA Assay Kit. The library was constructed using Illumina’s TruSeq Nano DNA LT Library Prep Kit. Agilent Bioanalyzer 2100 and Promega QuantiFluor were utilized to assess library quality. Raw sequencing data were in FASTQ format. 250 bp paired-end reads were generated and preprocessed by cutadapt software. Clean sequence reads were imported into QIIME2 (Bolyen et al., 2019), and variant calling was carried out using DADA2 (Callahan et al., 2016). The amplicon sequence variant (ASV) was clustered based on 100% similarity. Species annotation was performed using a pre-trained Naive Bayes classifier aligned with the SILVA 138 reference database.

To quantify the changes in gut microbiota after the intervention, we compared the ASV profiles of the three groups. The QIIME2 software was used to analyze alpha and beta diversity (https://docs.qiime2.org). Alpha-diversity indexes (ACE, Chao1, Shannon, and Simpson) were estimated. Principal coordinates analysis (PCoA) based on unweighted UniFrac distances was then used to assess beta diversity (Lozupone et al., 2007), showing the similarity of the microbial community. Differential abundance analysis of gut microbiota was performed through the Kruskal-Wallis test. To identify the representative taxa of each group, linear discriminant analysis effect size (LEfSe) was used to detect different features among the three groups (Segata et al., 2011). LDA > 4 and p < 0.05 were set as cutoff values to define significantly different genera.

Statistical analysis

Baseline differences were evaluated using one-way ANOVA. Tukey post hoc test was used for comparisons between more than two groups. Data that did not satisfy the normal distribution were compared using non-parametric tests, and differences between groups were compared using Kruskal-Wallis. Repeated-measures analysis of variance (ANOVA) was used to analyze spatial MWM data. Spearman’s correlation analysis was performed to explore the relationship between inflammatory markers or cognitive function and gut microbiota composition. The SPSS software package 26.0 (IBM Corporation, Armonk, NY, USA) and R, version 4.1.0 (R Core Team, 2021) were used for the above analyses. LEfSe analysis used the non-parametric factorial Kruskal-Wallis rank-sum test (Kruskal & Wallis, 1952) and the (unpaired) Wilcoxon rank-sum test (Wilcoxon, 1945; Mann & Whitney, 1947) in combination with liner discriminant analysis (LDA) (Fisher, 1936) effect sizes to find robust differential species between groups. Figures were drawn using Graph Pad Prism 8.0 software. A value of p < 0.05 was regarded as a significant difference.

Results

Semaglutide decreased body weight gain and ameliorated inflammatory markers

Inflammation is a prevalent process in obesity. At the end of the experiment, HFD induced an increase in body weight (Fig. 1A). Levels of inflammatory markers levels, including TNF α, IL-6, and IL-1β were elevated in the HFD group compared to the NCD group (Figs. 1B–1D). However, semaglutide treatment significantly reduced body weight and levels of inflammatory markers (all p < 0.05).

Figure 1 Body weight and inflammatory markers in the three groups of mice.

(A) Body weight. (B) Tumour necrosis factor α (TNF α) levels. (C) Interleukin-6 (IL-6) levels (D) Interleukin-I β (L-1 β) levels. Results are shown as box plots (line, median; box, interquartile range (IQR); whiskers, 1.5 × IQR), n = 6 mice for each group. An asterisk (*) indicates p < 0.05 compared with NCD and a number sign (#) indicates p < 0.05 compared with HFD.

Semaglutide improved cognitive function

To explore the effects of semaglutide treatment on cognitive function, the MWM test was used to examine learning and memory function. During the 5-day learning phase, escape latencies were significantly shorter in the Sema group than in the HFD group (Fig. 2A). On the sixth day of the probe test period, the HFD group showed decreased time spent in the target quadrant (TSTQ) and the number of times crossing the platform area (NTCPA) compared with the NCD group (all p < 0.05). In contrast, TSTQ and NTCPA were greater in the Sema group than in the HFD group (all p < 0.05). There was no statistical difference between the three groups in total swimming distance (TSD) and average swimming speed (ASP) within 60 s (Figs. 2B–2E).

Figure 2 Semaglutide improve cognitive decline in obese mice. The morris water maze (MWM) test was perforemed to evaluate the cognitive function.

(A) Escape latency from day 1 to 5. (B) Percentage of time spent in the target quadrant (TSTQ) within 1 min at day 6. (C) The number times of crossing the platform area (NTCPA) within 1 min at day 6. (D) Total swimming distance (TSD) within 1 min at day 6. (E) Average swimming speed (ASP) within 1 min at day 6. Results are shown as box plots (line, median; box, interquartile range (IQR); whiskers, 1.5 × IQR), n = 8 mice for each group. An asterisk (*) indicates p < 0.05 compared with NCD and a number sign (#) indicates p < 0.05 compared with HFD.

Alterations of gut microbiota diversity associated with semaglutide treatment

The microbiota was analyzed by 16S rRNA gene sequencing. The amplicon sequence variant (ASV) rank abundance curve was based on ASV serial number and relative abundance as axes to draw a hierarchical clustering curve. The curve was wide and smooth in the horizontal direction, indicating that the evenness and richness of the three groups were good (Fig. 3A). As shown in the Venn diagram, the total number of ASV was 2632, and the number of shared ASV was 211. In addition, 922, 544, and 654 ASV were unique to the NCD, HFD and Sema groups, respectively (Fig. 3B). For the diversity index analysis, abundance-based coverage estimators (ACE) was significantly higher in the Sema group than in the HFD group (p < 0.05). Compared with the NCD group, the Chao1, Shannon, and Simpson indices were also reduced in the HFD group (p < 0.05), while there was no significant difference in the values between the Sema group and HFD groups (Figs. 4A–4D). The Principal Coordinate Analysis (PCoA) based on unweighted UniFrac distances (Fig. 4E) showed that the bacterial composition was significantly different among the three groups (ANOSIM, unweighted UniFrac R = 0.816, P = 0.001). The PCoA plots also showed that the HFD and Sema groups were distinctly separated, indicating that the semaglutide treatment significantly affects intestinal flora diversity in obese mice.

Figure 3 ASV rank abundance (A) and Venn diagram (B) of the NCD group, HFD group and Sema group.

Figure 4 The effect of semaglutide on the gut microbiome diversity in obese mice.

Fecal microbiome composition was analyzed by 16S rRNA gene sequencing (n = 6). (A) ACE index. (B) Chao 1 index. (C) Shannon index. (D) Simpson index. (E) Principal coordinate analysis (PCoA) analysis of fecal microbiota based on unweighted UniFrac distance. The distinct clustering of samples was observed. The percentage of variation explained by PCoA1 and PCoA2 are noted in the axes. Groups are distinguished by colors. Each colored symbol corresponds to an individual sample. An asterisk (*) indicates p < 0.05 compared with NCD and a number sign (#) indicates p < 0.05 compared with HFD.

Alterations in gut microbiota composition of mice treated with Semaglutide

To determine the effects of semaglutide on the microbial community, we compared the microbial taxa at the phylum, order, family, and genus level among three groups (Figs. 5A–5D).

Figure 5 The difference of gut microbiota among three groups.

(A) The composition of gut microbiota at the phylum level. (B) The composition of gut microbiota at the class level. (C) the composition of gut microbiota at the family level. (D) The composition of gut microbiota at the genus level. (E) Linear discriminant analysis (LDA) effect size (LEfSe) showing the most significantly abundant taxa enriched in microbiome from the three groups. The color of the histogram represents the respective group, and the length represents the LDA score, that is, the impact of significantly different taxa between different groups. It mainly shows the significantly different taxa with LDA score > 4; p < 0.05 for Kruskal-Wallis test.

At the phylum level, compared with the NCD group, the relative abundance of Firmicutes and Desulfobacterota was higher, while the relative abundance of Verrucomicrobiota and Proteobacteria was lower in the HFD group. Compared with the HFD group, the relative abundance of Firmicutes and Desulfobacterota was decreased, while the relative abundance of Verrucomicrobiota and Proteobacteria was increased in the Sema group (Fig. 5A).

At the class level, the relative abundance of Bacilli, Clostridia, and Desulfovibrionia was higher in the HFD group compared to the NCD group, while the relative abundance of Verrucomicrobiae and Gammaproteobacteria was lower in the HFD group. Compared with the HFD group, the relative abundance of Bacilli, Clostridia, and Desulfovibrionia was decreased while the relative abundance of Verrucomicrobiae and Gammaproteobacteria was increased in the Sema group (Fig. 5B).

At the family level, compared with the NCD group, the relative abundance of Erysipelotrichaceae, Lachnospiraceae, Desulfovibrionaceae and Peptostreptococcaceae was higher, while the relative abundance of Lactobacillaceae, Muribaculaceae and Akkermansiaceae was lower in the HFD group. Compared with the HFD group, the relative abundance of Erysipelotrichaceae, Lachnospiraceae, Desulfovibrionaceae and Peptostreptococcaceae were decreased while the relative abundance of Lactobacillaceae, Muribaculaceae and Akkermansiaceae was increased in the Sema group (Fig. 5C).

At the genus level, compared with the NCD group, the relative abundance of Dubosiella, Romboutsia and Odoribacter was higher, while the relative abundance of Muribaculaceae, Lactobacillus, Akkermansia was lower in the HFD group. Compared with the HFD group, the relative abundance of Dubosiella, Romboutsia, and Odoribacter was decreased, while the relative abundance of Muribaculaceae, Lactobacillus, and Akkermansia was increased in the Sema group (Fig. 5D).

Additionally, the linear discriminant analysis effect size (LEfSe) analysis (α = 0.05, linear discriminant analysis (LDA) score >4.0) was applied to display the species composition and differences of the samples visually (Fig. 5E). The HFD group displayed a significant increase in the abundance of Firmicutes at the phylum level, Bacilli at the class level, Erysipelotrichales and Peptostreptococcales_Tissierellales at the order level, Erysipelotrichaceae, Peptostreptococcaceae and Eggerthellaceae at the family level, Romboutsia, Dubosiella and Enterorhabdus at the genus level. The Sema group was characterized by Actinobacteriota and Verrucomicrobiota phylum. Additionally, the Sema group was enriched in Coriobacteriia and Verrucomicrobiae at the class level, Coriobacteriales and Verrucomicrobiales at the order level, Atopobiaceae and Akkermansiaceae at the family level, and Akkermansia and Coriobacteriaceae _UCG_002, Clostridia _UCG_014 at the genus level.

Relationship between gut microbiota composition and inflammatory markers

As shown in Fig. 6, Enterorhabdus (r =0.549), Romboutsia (r =0.482) and Dubosiella (r =0.501) were positively correlated with TNF α, while Clostridia _UCG_014 (r =  − 0.552) and Muribaculaceae (r =  − 0.495) were negatively correlated with TNF α (all p < 0.05). Romboutsia (r =0.568) was positively correlated with IL-6, whereas Clostridia _UCG_014 (r =  − 0.575) and Muribaculaceae (r =  − 0.485) were negatively correlated with IL-6 (all p < 0.05). Enterorhabdus (r =0.656, p < 0.01) and Romboutsia (r =0.550, p < 0.05) were positively correlated with IL-1β, whereas Clostridia _UCG_014 (r =  − 0.735, p < 0.01) was negatively correlated with IL-1β. Interestingly, these negatively related genera were deficient in the intestine of mice in the HFD group but enriched in the Sema group.

Figure 6 The relationship between microbiota composition and inflammatory parameters.

The colors range from blue (negative correlation) to red (positive correlation). An asterisk (*) indicates p < 0.05 and two asterisks (**) indicate p < 0.01.

Relationship between gut microbiota composition and cognitive function

As shown in Fig. 7, TSTQ and NTCPA were positively correlated with Muribaculaceae (r = 0.537, 0.497, respectively; all p < 0.05) and Clostridia _UCG_014 (r = 0.655, 0.595, respectively; all p < 0.01). TSTQ and NTCPA were negatively correlated with Romboutsia (r =  − 0.566, −0.560, respectively; all p < 0.05) and Dubosiella (r =  − 0.548,−0.671, respectively; all p < 0.05). TSD was negatively correlated with Enterorhabdus (r =  − 0.492; p < 0.05). Interestingly, these positively related strains were enriched in the intestine of mice in the Sema group, but deficient in the HFD group.

Figure 7 The relationship between microbiota composition and cognitive function.

The relationship between microbiota composition and cognitive function. The colors range from blue (negative correlation) to red (positive correlation). An asterisk (*) indicates p < 0.05 and two asterisks (**) indicate p < 0.01.

Discussion

Our study provides a novel perspective that semaglutide may play an important role in modulating gut microbiota composition against cognitive impairment and inflammation.

GLP-1 is a gut-derived peptide produced by intestinal epithelial L-cells in response to fat and carbohydrate intake (Brown et al., 2021). The neuroprotective effects of GLP-1 are mediated by modulating learning and memory (Müller et al., 2019), decreasing inflammation and apoptosis (Diz-Chaves et al., 2024; During et al., 2003) and promoting the level of nerve growth factor (Perry et al., 2002). GLP-1 receptor agonists include exenatide, liraglutide, lixisenatide, dulaglutide, and semaglutide (Madsbad, 2016). Semaglutide is available for the treatment of overweight and obesity in people with or without type 2 diabetes (Kadowaki et al., 2022). Recent research suggests that semaglutide is effective in lowering body weight, improving glycemic control and decreasing cardiometabolic risk factors in people with type 2 diabetes. However, no reports have been published on the effects of semaglutide on obesity-related gut microbiota. The microbiota plays a key role in the host’s digestion, metabolism, and behavior, but whether semaglutide could help increase the beneficial gut microbiota of HFD-fed mice, which might impact the improvement of cognitive function and inflammation, is poorly understood.

PCoA showed that the Sema group and the HFD group were clustered into two categories, indicating that semaglutide could affect the intestinal flora diversity of mice. We also found significant differences in the relative abundance of Firmicutes and Bacteroidota in the NCD and HFD groups. The HFD can significantly increase the abundance of Firmicutes and decrease the relative abundance of Bacteroidota, resulting in gut microbiota dysbiosis (Zhao et al., 2022). After semaglutide intervention, the relative abundance of Firmicutes was dramatically reduced and the proportion of Bacteroidota was not greatly augmented. To further evaluate the precise changes in the gut microbiota, we analyzed the microbiota differences at the order, family, and genus levels by comparing the histogram of species classification. Administration of HFD led to a significant elevation in the relative abundance of Romboutsia, Dubosiell a, Enterorhabdus and decrease of Akkermansia, Muribaculaceae, Coriobacteriaceae _UCG_002, Clostridia _UCG_014, whereas intervention of semaglutide could dramatically reverse the relative abundance of these groups. Previous reports have also shown that HFD may decrease the relative abundance of Muribaculaceae in animals and individuals (Ye et al., 2021).

The rapid increase in the level of Akkermansia was the most interesting finding in this study. We observed that Akkermansia in the Sema group was nearly 166 times as high as in the HFD group, showing the largest change in the proportion of the whole genus. Akkermansia first isolated by Derrien et al. (2008) is a nonmotile and strictly anaerobic Gram-negative bacterium with about 1–4% abundance in the gut. Akkermansia muciniphila has been reported to have many beneficial effects, including reducing fat mass gain and glycemia (Derrien, Belzer & De Vos, 2017; Anhê, Schertzer & Marette, 2019), improving various metabolic abnormalities (Cani & Knauf, 2021) and alleviating neurodegenerative processes (Ou et al., 2020). A recent study demonstrated that a novel protein P9 secreted by Akkermansia muciniphila could promote GLP-1 secretion (Cani & Knauf, 2021). Porras et al. (2019) reported a negative correlation between the NAFLD activity score and the abundance of Akkermansia. Moreover, studies have suggested that a link exists between Akkermansia muciniphila and cognitive performance, and the underlying mechanism involves decreasing the level of pro-inflammatory cytokine interleukin (IL)-6 in both peripheral blood and the hippocampus (Zhu et al., 2023). We also found that semaglutide could significantly enhance cognitive performance and increase Akkermansia muciniphila levels in obese mice. However, there is no direct evidence to validate that transplantation of the gut microbiota of mice after semaglutide intervention could also display similar cognitive function. Furthermore, Akkermansia did not correlate significantly with inflammatory markers or cognitive function. This may be due to small sample size. More studies need to be further explored.

Muribaculaceae has been found to degrade a variety of complex carbohydrates (Lagkouvardos et al., 2019) and is increased in response to a high-amylose maize-resistant starch diet (Barouei et al., 2017). Members of this family could increase the production of succinate, acetate and propionate (Smith et al., 2019) and decrease fat absorption or deposition (du Preez et al., 2021) which has been linked to longevity in rodents (Sibai et al., 2020) or humans (Li et al., 2016). In this study, the prevalence of Muribaculaceae was markedly increased by semaglutide, as the increased relative abundance of Muribaculaceae was positively related to cognitive function.

Being an obesity-related phylotype (Tu et al., 2020), Romboutsia was significantly associated with lipid profile and lipogenesis in the liver (Do et al., 2020). Previous research has found that a high-fat/high-sugar diet could enhance the abundance of harmful genera (Romboutsi a, Clostridium) and reduce the abundance of beneficial probiotic genera (Bifidobacterium, Lachnospiraceae-NK4A136, Ileibacterium) (Wang et al., 2021). Our findings are consistent with the above results (Yin et al., 2023; Fu et al., 2021). Additionally, semaglutide decreased the abundance of Romboutsia, which was positively correlated with IL-6 and IL-1β. IL-6 has been identified as a blood marker of cognitive decline and severity of cognitive impairment (Di Benedetto et al., 2017; Trapero & Cauli, 2014). Marsland et al. (2008) have also reported an inverse association between IL-6 and memory function in mid-life adults.

Previous studies have reflected significant negative correlations between obesity-related indexes and Dubosiella, suggesting that Dubosiella might inhibit obesity (Guo et al., 2021). However, in our study, the abundance of Dubosiella was increased in the HFD group. Another study has found that the Erysipelotrichaceae members Dubosiella and the Eggerthellaceae member Enterorhabdus were positively correlated with obesity-related parameters (He et al., 2022). More research should be carried out in this field to assess this microorganism’s behavioral patterns fully.

There are some limitations to our study. First, the gut microbiomes were only profiled by 16S rRNA gene sequencing. Untargeted metabolomic analysis of serum and feces and fecal microbiota transplantation (FMT) intervention also needs further evaluation. Whether transplantation of the gut microbiota of mice after semaglutide intervention into HFD-fed mice could show improvement in cognitive function needs to be further investigated. Second, the semaglutide intervention was administered for only 12 weeks, which is relatively short. Third, no other behavioral tests were included to assess different cognitive domains. Therefore, more studies are warranted to address these issues.

Conclusions

In summary, we discovered that semaglutide significantly increased the relative abundance of Akkermansia, Muribaculaceae, Coriobacteriaceae _UCG_002, and Clostridia _UCG_014 at the genus level and decreased the relative abundance of Romboutsia, Dubosiella, and Enterorhabdus. This supports the idea that semaglutide can regulate the intestinal flora disorder caused by a high-fat diet. These beneficial bacteria may effectively improve cognitive function in obese mice by reducing IL-6 and IL-1β production.

Supplemental Information

Supplemental Information 1 Data for Fig. 1

Body weight and inflammatory markers in the three groups of mice (n = 6)

Supplemental Information 2 Data for Fig. 2

Semaglutide improve cognitive function (n = 8). (A) The escape latency from the day 1 to day 5. (B) Time spent in the target quadrant (TSTQ) within 1 min at the day 6. (C) The number times of crossing the platform area (NTCPA) within 1 min at the day 6. (D) Total swimming distance (TSD) within 1 min at the day 6. (E) Average swimming speed (ASP) within 1 min at the day 6. n = 8 per group. * p < 0.05 vs NCD and # p < 0.05 vs HFD.

Supplemental Information 3 Data for Fig. 3

Venn diagram of the NCD group, HFD group and Sema group.

Supplemental Information 4 Data for Fig. 3

Venn diagram of the NCD group, HFD group and Sema group.

Supplemental Information 5 Data for Fig. 3

Venn diagram of the NCD group, HFD group and Sema group.

Supplemental Information 6 Data for Fig. 3

Venn diagram of the NCD group, HFD group and Sema group.

Supplemental Information 7 Data for Fig. 4

The α-diversity and β-diversity of gut microbiota in the three groups. n = 6 per group

Supplemental Information 8 Data for Fig. 5

The composition of gut microbiota in the three groups.

Supplemental Information 9 Code for Figs. 6 and 7

Supplemental Information 10 The correlation coefficient and the statistical results of Fig. 6

Supplemental Information 11 The correlation coefficient and the statistical results of Fig. 7

Supplemental Information 12 Cladogram of the most differentially abundant taxa in the NCD group, HFD group and Sema group

Different colors indicate different groups. Nodes of different colors indicate the microbes that play an important role in the group represented by the color. From the inside to the outside, each circle is the species at the level of phylum, class, order, family, and genus.

Supplemental Information 13 Comparisons of the relative abundance at the genus level in the three groups by STAMP analysis

Supplemental Information 14 ARRIVE 2.0 Checklist

Supplemental Information 15 NCD_1_R1 sequence data

Supplemental Information 16 NCD_1_R2 sequence data

Supplemental Information 17 NCD_2_R1 sequence data

Supplemental Information 18 NCD_2_R2 sequence data

Supplemental Information 19 NCD_3_R1 sequence data

Supplemental Information 20 NCD_3_R2 sequence data

Supplemental Information 21 NCD_4_R1 sequence data

Supplemental Information 22 NCD_4_R2 sequence data

Supplemental Information 23 NCD_5_R1 sequence data

Supplemental Information 24 NCD_5_R2 sequence data

Supplemental Information 25 NCD_6_R1 sequence data

Supplemental Information 26 NCD_6_R2 sequence data

Supplemental Information 27 HFD_1_R1 sequence data

Supplemental Information 28 HFD_1_R2 sequence data

Supplemental Information 29 HFD_2_R1 sequence data

Supplemental Information 30 HFD_2_R2 sequence data

Supplemental Information 31 HFD_3_R1 sequence data

Supplemental Information 32 HFD_3_R2 sequence data

Supplemental Information 33 HFD_4_R1 sequence data

Supplemental Information 34 HFD_4_R2 sequence data

Supplemental Information 35 HFD_5_R1 sequence data

Supplemental Information 36 HFD_5_R2 sequence data

Supplemental Information 37 HFD_6_R1 sequence data

Supplemental Information 38 HFD_6_R2 sequence data

Supplemental Information 39 Sema_1_R1 sequence data

Supplemental Information 40 Sema_1_R2 sequence data

Supplemental Information 41 Sema_2_R1 sequence data

Supplemental Information 42 Sema_2_R2 sequence data

Supplemental Information 43 Sema_3_R1 sequence data

Supplemental Information 44 Sema_3_R2 sequence data

Supplemental Information 45 Sema_4_R1 sequence data

Supplemental Information 46 Sema_4_R2 sequence data

Supplemental Information 47 Sema_5_R1 sequence data

Supplemental Information 48 Sema_5_R2 sequence data

Supplemental Information 49 Sema_6_R1 sequence data

Supplemental Information 50 Sema_6_R2 sequence data

We thank the reviewers whose constructive evaluation and suggestions have helped us improve the quality of this manuscript.

Additional Information and Declarations

Competing Interests

Author Contributions

Animal Ethics

DNA Deposition

Data Availability

The authors declare there are no competing interests.

Jing Feng performed the experiments, analyzed the data, prepared figures and/or tables, and approved the final draft.

Zhenjie Teng analyzed the data, prepared figures and/or tables, and approved the final draft.

Yu Yang analyzed the data, prepared figures and/or tables, and approved the final draft.

Jingzhen Liu analyzed the data, prepared figures and/or tables, and approved the final draft.

Shuchun Chen conceived and designed the experiments, authored or reviewed drafts of the article, and approved the final draft.

The following information was supplied relating to ethical approvals (i.e., approving body and any reference numbers):

This study was approved by Animal Ethics Committee of Hebei General Hospital (202332).

The following information was supplied regarding the deposition of DNA sequences:

The raw reads are available at NCBI Sequence Read Archive (SRA): PRJNA1074437.

The following information was supplied regarding data availability:

The raw measurements are available in the Supplemental Files.

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
