# Peer review of "Effects of semaglutide on gut microbiota, cognitive function and inflammation in obese mice"

_PeerJ, doi:10.7717/peerj.17891_

## Round 0.1 · original submission · Major Revisions

Please submit a revised version after carefully considering the extensive reviews and helpful suggestions provided by the reviewers.

Reviewer 1 ·

Basic reporting

This manuscript investigates the effect of semaglutide on gut microbiota modulation and its potential link to improved cognitive function. The authors present two main experiments:

1. Cognitive Assessment: The Morris water maze test was used to evaluate spatial learning and memory.
2. Microbiome Analysis: 16S rRNA sequencing was performed to characterize changes in gut microbial composition.

General Recommendations:
The study design and findings are presented clearly. Proofreading for typos, spacing, and misspellings will further enhance readability.

Figures:
1. Readability: Increase font sizes in Figures 3, 5, and 4e. Improve overall image resolution for publication.
2. Box Plots: Consider using box plots in Figure 2 to provide a clearer visual representation of the distribution within each sample group.
3. LEfSe Analysis: Separate differentially abundant taxa by rank (in subplots or within the same plot) to enhance visualization and interpretation.
4. Cladogram (Figure 5b): If the cladogram doesn't significantly enhance the narrative and it's difficult to interpret, consider removing it or placing it in supplementary materials.
5. Taxonomic Classification: Include additional figures depicting taxonomic classification at various levels (phylum, class, order, family, genus, species) to highlight differences between groups and provide insights for future research.
6. Correlations: Illustrate individual samples in a linear regression to identify potential outliers or unique patterns within the data.

Experimental design

For complete transparency and to enable future analysis, please provide the following:

* QIIME 2 Parameters:
1. Quality Filtering: Specify the method (e.g., DADA2, Deblur) and the exact quality filtering parameters used (e.g., truncation lengths, quality thresholds).
2. ASV Creation: Describe the process of generating amplicon sequence variants (ASVs) and detail any relevant parameters employed within DADA2 or Deblur.
3. Database: Name the specific reference database used for taxonomic assignment (e.g., SILVA 138, Greengenes 13_8).

* LEfSe Analysis Parameters:
1. Statistical Test: Identify the statistical test used for biomarker discovery (e.g., Kruskal-Wallis test, Wilcoxon rank-sum test).
2. Effect Size Threshold: State the LDA (Linear Discriminant Analysis) score threshold used to determine significance.
3. Other Relevant Settings: Mention any additional parameters that were important to your LEfSe analysis.

* Code Availability: Share the code used specifically for Spearman correlations on a suitable platform such as GitHub, Bitbucket, or GitLab. This promotes reproducibility.

* Sequence Deposition: Deposit the raw 16S rRNA sequences in a public repository like NCBI, EMBL-EBI, or DDBJ. Include the accession numbers in the manuscript, allowing future researchers to access and analyze the data.

* Beta Diversity Metric: Please specify the exact metric used to calculate beta diversity (e.g., Bray-Curtis dissimilarity, weighted UniFrac, unweighted UniFrac).

Validity of the findings

* The authors mention that Semaglutide treatment could contribute to increased richness and diversity. However, the provided information suggests that while richness may be similar to the NCD group, diversity might differ. Please clarify this statement or provide the missing information to support the conclusion about increased diversity.

* In the Venn diagram (Figure 3B), please specify how much of the total abundance per group is represented by the shared ASVs in each section. For example, what percentage of the NCD group's total abundance is represented by the 211 shared ASVs? Additionally, plotting Venn diagrams at different taxonomic levels (phylum, class, order, family, genus, and species) would provide a more nuanced view of the shifts in microbial composition.

* The authors suggest that Semaglutide treatment could re-establish a 'normal' gut microbiota. Can you quantify how different the Semaglutide-treated group's microbiome is from the NCD group in terms of abundance? The provided evidence indicates changes in the microbiome, but it might be more accurate to describe this as a change to a different state rather than a complete restoration.

The experimental design was well performed and evaluated. Nevertheless, to strengthen the conclusion about improved cognitive function, consider combining the MWM with other behavioral tests assessing different cognitive domains. This would provide a more comprehensive picture. While the Morris Water Maze is a valuable tool for studying spatial learning and memory, interpreting the results in the context of its limitations is crucial.

Additional comments

Addressing these recommendations will strengthen the manuscript and contribute to its successful publication.

Reviewer 2 ·

Basic reporting

In general, this manuscript reads as a draft version, and not as a publication-quality polished article. Authors need to revise their article sentence by sentence to improve the precision of the wording, and remove typos and grammar mistakes. Also, figures should be polished and improved to reach publication quality. Figure captions should have more detail on what is presented.

Statistical analysis. In the corresponding methods section, it is stated that one-way ANOVA was used, with subsequent Tukey’s or Dunnett’s tests. It is not clear, however, which of these two post hoc tests was used in each case. Please indicate in this information in the captions of figures 1, 2 and 4. Moreover, please provide the p-values of all the global ANOVA tests as well. In many cases, there appears to be large overlaps between the error-bars of groups which are marked as significantly different, but no explanation is given as what these intervals represent. Please use 95% confidence intervals, so that significant differences can be appreciated as separated error bars.

Figure 3. The colors are not easy to distinguish, and it is difficult to see which curves correspond to each treatment. I suggest using only three colors, one for each treatment.

Line 170. "...showing excellent abundance and evenness..." Please avoid adjectives like "excellent" to describe your results. Use objective descriptions instead.

Lines 178-179: The PCoA results could be explained with more detail. It seems like the first principal coordinate separates the samples based on diet, while the second principal coordinate separates them according to the administration of semaglutide.

Figures 3, 4 and 5. Please use a readable font size for all text in all figures. In general, it is advisable to re-plot all figures with publication quality. For instance, taxonomic names in figure axes should be properly formatted, including the use of italics for genus and species names, and avoiding underscore characters in names. For example, instead of f_Eggerthellaceae write Eggerthellaceae (f).

Figure 5a. Are these average relative abundance of the 8 samples for each treatment? Please explain in the figure's caption.

Figures 5a and 5c look interesting, but they cannot be understood, as the taxonomic labels are unreadable. As indicated, please use larger font sizes or figures with higher resolution.

Lines 187-188: "After 12 weeks of semaglutide administration, the relative abundance of the genus was dramatically decreased." Of which genus?

Line 242: "...semaglutide could dramatically reverse the relative abundance of these families." Since the mentioned groups belong to different taxonomic levels, I suggest: "...semaglutide could dramatically reverse the relative abundance of these groups."

Lines 288-289: "Metagenomic and metabonomics approaches are needed to further
evaluate." Please elaborate.

Sequencing reads do not seem to be available in public databases.

Experimental design

The bioinformatic analysis should be explained with more detail, including all important steps. For example, there is no explanation of how the taxonomic assignment of amplicon sequences was made. Furthermore, citations of all the methods used for bioinformatics are lacking. As mentioned, the original amplicon reads do not seem to be publicly available.

Validity of the findings

The causal relationships between the different variables evaluated in this study is not clear, and the limitations in this regard should be made explicit in the wording throughout the whole manuscript. From the experimental design and the obtained results, one can conclude that semaglutide decreases obesity, inflammation markers and cognitive impairment under a high-fat diet. Moreover, semaglutide under high fat results in shifts in microbial communities, increasing the relative abundance of known beneficial microorganisms. These are clear and valuable results on their own, and have in my opinion enough merit for publication. However, the causal relationships between these response variables cannot be inferred from the experimental design. In particular, whether the changes in the microbial communities cause the alleviation of obesity, inflammation markers and cognitive decline cannot be inferred from the current experimental design. The correlations shown in figures 6 and 7 could arise from causal relations, but they could also be the result of effects semaglutide administration on both members of the variable-pairs (i.e., they could be spurious). To distinguish between these possibilities would require additional experiments, and is in my opinion outside of the scope of this article. Thus, the authors should remove any statements affirming a causal link between changes in the gut microbial communities and the physiologica/cognitive responses of obese mice to to semaglutide. This would include changes in most sections of the manuscript, including the title.

The handling and presentation of changes of particular taxonomic groups of bacteria in response to semaglutide is suboptimal, in my opinion. While it is interesting to see which taxa have the strongest effects in the separation of experimental groups in terms of beta diversity, figure 5c is not enough to appreciate the differences of these bacteria in the three experimental groups. A bar plot showing the relative abundance of these taxons (with confidence intervals) for each experimental group would be very informative. However, I also have reservations about the use of LEfSe as the sole method to identify responding ASVs, since it been found to have an inadequate false discovery rate (Nearing et al. 2022. Nat Commun 13, 342). Please consider using another method, preferably one that tests differences between experimental groups explicitly, of follow Nearing et al. advice and use a consensus approach.

Additional comments

The effect of semaglutide on the gut microbiome of model animals is of great interest to understand the interactions of this drug with overall health. Therefore, this article has great potential for publication, once all the issues have been addressed.

Reviewer 3 ·

Basic reporting

General comments:
The manuscript entitled 'Gut microbiota mediates positive effects of semaglutide on cognitive function in obese mice' investigates whether semaglutide could improve cognitive function by modulating the gut microbiota in obese mice. While the manuscript is well-written and most analyses are conducted properly, there are still many areas that need to be addressed before it can be published.

Experimental design

Major comments:
1. For statistical analysis, the Kruskal-Wallis test should be used instead of the ANOVA test, and adjusted p-values < 0.05 should be considered significant differences.
2. For PCoA, the authors are advised to conduct permutational multivariate analysis of variance (PERMANOVA) to determine significant differences among groups.
3. Only 6 out of 8 mice in each group underwent 16S rRNA sequencing. The authors should report the selection process in the Methods section.
4. Both correlation coefficients and p-values should be reported in the text for Pearson correlation analysis.

Validity of the findings

1. An availability of data statement is not provided in the manuscript. All raw reads data should be deposited in NCBI, and the accession number should be reported in the manuscript.

Additional comments

Minor comments:
1. In the abstract, the authors stated that 'Correlation analysis suggested that cognitive function was positively correlated with certain beneficial bacteria and negatively associated with harmful bacteria.' It is recommended to specify the particular microbes rather than using general terms like 'beneficial' and 'harmful' bacteria.
2. In lines 212-213: 'Interestingly, these negatively related taxa were enriched in the intestines of mice in the Sema group but deficient in the HFD group.' Since the taxa are at the genus level rather than the strain level, the authors should use 'genus' instead of 'strains' here.
3. In line 255: The lower case should be used for the first letter of ‘Studies’.
4. In line 277: There is a citation problem.

·

Basic reporting

Feng et al. reported the difference in inflammation, cognitive functions and intestinal microbiota between three groups of mice: a normal-chow diet group (NCD), high-fat diet (HFD) and HFD+semaglutide (Sema). The study tried to connect specific taxa enriched in Sema group with cognitive functions of the mice to shed a light on the effects of semaglutide on the improvement of cognitive function in obese mice. However, the study hasn’t been designed to directly connect the cognitive function with mice gut microbiota. For example, by removing specific taxa, the cognitive function is systematically impacted. Or the metabolites from specific taxa could enhance the cognitive function by some metabolism pathways. Based on this, please tune down the statement or conclusion throughout the manuscript, for example, ‘Semaglutide could prevent congnitive impairment by modifying the structure and composition of gut microbiota in HFD-fed mice (lines 38-39)’, as well as the title and lines 77-79, 229-230. Semaglutide could help increase the beneficial gut microtiota of HFD-fed mice, which might have impacts on the improvement of cognitive function.

The study also detected the correlations between specific taxa and inflammatory markers, and included some discussion on that. I would suggest to indicate this part in title, abstract, introduction and conclusion.

The study didn’t report the specific taxa enriched in NCD, relative to HFD. If these taxa are the same as those enriched in Sema, semaglutide may be able to restore the intestinal taxa of HFD-fed mice. Based on PCA analysis and the taxa composition in Figure 5A, the microbial community patterns differed between NCD and Sema. So please be careful when stating that semaglutide could restore the intestinal taxa disorder such as lines 293-297.

Although this study is mainly to compare the intestinal taxa between HFD and Sema, I would suggest to include the specific taxa enriched in NCD relative to HFD, and discuss if any of these taxa is restored after semaglutide treatment. The results could be implemented in lines 182-196 and the discussion could be implemented in appropriated place.

In Lines 189-196, please describe what specific taxa enriched in each of three groups, NCD, HFD and Sema, from Fig 5. In lines 198-213, please explain any of these specific taxa showed significant correlations.

For Fig 5b, the same color means different things. For example, green color is NCD, and as well as f_Muribaculaceae. Does it mean f_Muribaculaceae is enriched in NCD? In the main text, f_Muribaculaceae is enriched in Sema which is blue color. What is the filled circle? What are the layers of circle with gradients of green, red or blue color? Please explain Fig 5b. The whole Fig 5 is to show what specific taxa are enriched in each group, NCD, HFD or Sema. Please use appropriate plots to show.

Fig 6 and 7 used hierarchical clustering analysis, which is not necessary. I would suggest to use the same order of taxa for the vertical for both fig 6 and 7. Either the taxa order of fig 6 or 7. With this way, it is easy to compare the correlations for each taxa between fig 6 and 7. For example, Clostridia_UCG_014 is on the top and shows the correlations with inflammatory markers in Fig 6 while at the same time this taxon is also on the top of Fig 7 and shows the correlations with cognitive functions.

Line 48: delete ‘has’
Line 105: please check ‘removal eyeball’
Lines 114-115: delete ‘was’. Please rephrase this sentence.
Line 133: change ‘calculated’ to ‘used’
Lines 204-206 and 212-213, how about positive correlations?
Line 212: Please check fig 7. ‘negative’ should be ‘positive’?
Line 232: change ‘fora’ to ‘flora’
Line 235: put a comma before ‘resulting in gut…’. Please also include citations here if these taxa could result in gut microbiota dysbiosis.
Lines 245-262: please explain why Akkermansia didn’t show any significant correlations with inflammatory markers or cognitive functions.
Line 276: change ‘I1-6’ to ‘IL-6’.

Experimental design

In Methods, please explain what specific primers were used in this study (Line 126) and the details on how to analyze the raw sequences and thus yield the ASV table (Lines 130-131).

Validity of the findings

The study also detected the correlations between specific taxa and inflammatory markers, and included some discussion on that. I would suggest to indicate this part in title, abstract, introduction and conclusion.

---

## Round 0.2 · Minor Revisions

Kindly ensure that these remaining minor yet abundant comments and details are carefully addressed at this stage.

Reviewer 1 ·

Basic reporting

I appreciate the authors' thorough attention to the suggestions and corrections provided by the reviewers in this revised version of the manuscript. The enhancements implemented have significantly elevated the overall quality of the article, and I commend the authors for their dedication. However, to ensure the successful publication of this manuscript, it is imperative to address the following minor suggestions:

1. Figure Presentation: In Figure 4E, there are unexplained groups marked as biomarkers without sufficient explanation, such as 'g__uncultured'. I recommend the authors provide relevant information for each group, for example, the family or order, or the closest assigned taxa name. For instance, "f_Vibrionaceae;g_uncultured" could be used.
2. Code Availability: To ensure the replicability of your results, I highly suggest uploading a file with the code used. While some R code files are present, a well-structured pipeline (e.g., a QIIME pipeline) would be very valuable for readers.
3. Sequencing Data and Metadata: The sequencing data and associated metadata should be deposited in a public database such as NCBI to ensure accessibility and reproducibility of the research findings.
4. Spelling and Readability: While this manuscript has improved in readability compared to its previous version, there are some misspellings that require attention. Additionally, I suggest double-checking the taxa names and ensuring that only the genus and species names are italicized.
5. Graph Quality: The quality of the final graphs should be improved prior to publication to ensure they are clear and professionally presented.

Experimental design

For future studies, I recommend including an additional control group receiving a standard chow diet alongside Semaglutide administration. This would allow for a more nuanced evaluation of Semaglutide's independent effects on cognitive function, separate from any dietary influences.

Validity of the findings

As suggested in the first review, expanding the cognitive function tests would further strengthen the findings and provide a more comprehensive assessment in relation to the proposed hypothesis.

Reviewer 2 ·

Basic reporting

I thank the authors for the corrections made regarding basic reporting, as the figures are much clearer now. The English language can still use improvement. Here are all the instances of typos, spelling or grammar that I could find:

Line 52: “A recent study has showed that…” should be “A recent study has shown that…”
Line 69: “An increasing body of evidences have…” should be “An increasing body of evidence has…”
Line 72. “A study has revealed that the Akkermansia muciniphila…” should be “A study has revealed that Akkermansia muciniphila…”
Lines 77-79 “There is evidence that liraglutide could regulate the composition of the gut microbiota in HFD-fed mice, especially Akkermansia (Zhao et al., 2022).” This phrease is not clear. What is the effect of liraglutide on the abundance of Akkermansia, found by Zhao et al., (2022)?
Line 84: “To investigate the underlying mechanisms of semaglutide, we use…” should be “To investigate the underlying mechanisms of semaglutide, we used…”
Line 91: “A total of 24 male SPF grade…” should be “A total of 24 male specific-pathogen-free grade…”
Line 93: “485*200*200” should be “485×200×200” Please use the multiplication symbol (similar to an x).
Line 94: “20-24çC” should be “20-24°C” Please use the correct symbol for degrees.
Line 100: “30nmol/kg/d” should be “30 nmol/kg/d” (Insert whitespace)
Line 106: “50mg/kg” should be “50 mg/kg” (Insert whitespace)
Line 112: “3000×g/min” There is an error in the units here.
Line 113: “4 °C” should be “4°C” Please remove the whitespace for consistency with other appearances of temperatures in the document.
Line 116: “Morris water maze (MWM) test” should be “Morris water maze test”
Line 117: “The Morris water maze test was…” should be “The Morris water maze (MWM) test was…” Please introduce the abbreviation here, and not in the subtitle.
Line 121, 124: “60s” should be “60 s”
Line 123: “10s” should be “10 s”
Line 129: “Gut Microbiota Analysis” should be “Gut microbiota analysis”
Line 136: “72 °C for 30 s. Finally 72 °C for 5 min.” should be “72°C for 30 s. Finally 72°C for 5 min.” (Remove 2 whitespaces).
Line 137: “and and purified” please remove one “and”
Line 140-141: “At last, 250bp paired-end reads were generated and preprocessed by cutadapt software.” “At last” is appropriate to describe the protocol, even if the authors waited a long time for their results. Also, a whitespace is missing again between a number and its corresponding units.
Line 153: “among three groups” should be “among the three groups”
Line 154: “genus” should be plural here: “genera”
Line 157: “All data were conducted using SPSS 26.0…” Something is missing here. Please be careful when correcting this, as all data analyses were not conducted using SPSS 26.0, since the previous section indicates that different software was also used for data analysis.
Lines 162-163: “LEfSe analysis was used Kruskal-Wallis test and Wilcoxon test in combination with liner discriminant analysis (LDA) effect sizes to find robust differential species between groups.” Something is missing here.
Line 170: “Inflammation is a prevalent event in obesity.” Please choose a word more suitable than “event”
Line 189: “As shown in venn figure,” should be “As shown in the Venn diagram,”
Line 195: “The Principle coordinate analysis” should be “The Principal Coordinate Analysis”
Line 326: “Marsland et al has also…” should be “Marsland et al. has also...”
Figure 2 caption: “The escape latency from the day 1 to day 5” should be “Escape latency from day 1 to 5”.
Figure 2 caption: “at the day 6” should be “at day 6” in all cases.

Experimental design

no comment

Validity of the findings

no comment

Additional comments

The manuscript has greatly improved, and I congratulate the authors for their effort. However, I do have some minor concerns:

Minor concerns

Figure 4b: Using the data provided by the authors, I was able to reproduce the Kruskal-Wallis test (Kruskal-Wallis chi-squared = 6.4678, df = 2, p-value = 0.0394). However, it was not clear how to reproduce the pair-wise comparisons. From the methods section: “Data that did not satisfy the normal distribution were compared using non-parametric tests, and differences between groups were compared using Kruskal-Wallis.” Doing another Kruskal-Wallis considering only the Sema and HDF groups, I get no sigificant differences (Kruskal-Wallis chi-squared = 1.2564, df = 1, p-value = 0.2623). However, the # symbol indicates a significant difference between these two groups. A Wilcoxon rank sum exact test also turns out not significant for this comparison (W = 11, p-value = 0.3095). Could the authors please clarify which test was applied, and the p-value for the comparison? Furthermore, could the authors please check whether this comparison is indeed significant?

Lines 129-154. Section “Gut Microbiota Analysis” lacks references for the bioinformatic methods used.

Lines 132-133: “Amplicon sequence variant (ASV) was clustered based on 100% similarity.” This is confusing, as it means that no clustering of OTUs was performed, but amplicon sequence variants were used instead.

Lines 187-189: “The amplicon sequence variant (ASV) rank abundance curve was wide and gently sloping, showing species abundance and evenness (Fig. 3A)” It is not clear from this description what these curves show about species abundance and evenness. Please be more specific in the description. Also, please describe the differences in these curves in the three mice groups.

Figure 3. The curves are very difficult to discern with so many colors. Please use one color per treatment, using the same color scheme as in Figure 3b. The fact that other published works use many colors in a similar plot does not make it readable or useful here. The individual mice numbers are of no consequence, but their experimental group is.

Reviewer 3 ·

Basic reporting

The authors have addressed most of the comments from reviewers. However, there are a few areas that should be made clear before the paper can be published.

Experimental design

For multiple comparisons, the p-values should be corrected. It’s unclear whether the authors reported the adjusted p-values or not.

Validity of the findings

The data availability statement should be put in the manuscript.

Additional comments

NA

·

Basic reporting

The authors addressed my concerns.

Experimental design

no comment

Validity of the findings

no comment

---

## Round 0.3 · Minor Revisions

Please consider the Editorial comments and the editorial suggestions in the attached file.

---

## Round 0.4 · accepted · Accept

Thanks for the improvements on the manuscript; your contribution is now accepted in PeerJ.